# Pulse-on-Demand Operation for Precise High-Speed UV Laser Microstructuring

**DOI:** 10.3390/mi14040843

**Published:** 2023-04-13

**Authors:** Jernej Jan Kočica, Jaka Mur, Julien Didierjean, Arnaud Guillossou, Julien Saby, Jaka Petelin, Girolamo Mincuzzi, Rok Petkovšek

**Affiliations:** 1Faculty of Mechanical Engineering, University of Ljubljana, Aškerčeva 6, SI-1000 Ljubljana, Slovenia; jernej.kocica@fs.uni-lj.si (J.J.K.); jaka.mur@fs.uni-lj.si (J.M.); jaka.petelin@fs.uni-lj.si (J.P.); 2Bloom Lasers, Avenue de Canteranne 11, 33600 Pessac, France; 3ALPhANOV, Aquitania Institute of Optics, Rue François Mitterrand, 33400 Talence, France

**Keywords:** pulse-on-demand, nanosecond pulses, UV laser, microstructuring

## Abstract

Laser microstructuring has been studied extensively in the last decades due to its versatile, contactless processing and outstanding precision and structure quality on a wide range of materials. A limitation of the approach has been identified in the utilization of high average laser powers, with scanner movement fundamentally limited by laws of inertia. In this work, we apply a nanosecond UV laser working in an intrinsic pulse-on-demand mode, ensuring maximal utilization of the fastest commercially available galvanometric scanners at scanning speeds from 0 to 20 m/s. The effects of high-frequency pulse-on-demand operation were analyzed in terms of processing speeds, ablation efficiency, resulting surface quality, repeatability, and precision of the approach. Additionally, laser pulse duration was varied in single-digit nanosecond pulse durations and applied to high throughput microstructuring. We studied the effects of scanning speed on pulse-on-demand operation, single- and multipass laser percussion drilling performance, surface structuring of sensitive materials, and ablation efficiency for pulse durations in the range of 1–4 ns. We confirmed the pulse-on-demand operation suitability for microstructuring for a range of frequencies from below 1 kHz to 1.0 MHz with 5 ns timing precision and identified the scanners as the limiting factor even at full utilization. The ablation efficiency was improved with longer pulse durations, but structure quality degraded.

## 1. Introduction

Laser microstructuring has been studied extensively in the last decades [1,2,3,4,5] due to its versatile, contactless processing and outstanding precision and structure quality [6] on a wide range of materials. The same is true for laser ablation using nanosecond pulses, as the corresponding sources are widely available. Numerous commonly used materials, including metals [7], exhibit low reflectivity in the ultraviolet (UV) spectrum; thus, nanosecond material processing in UV is advantageous compared to IR and green pulses [8,9]. The advantages are inherently tighter focusing, better efficiency, and lower heat-related impact on the surrounding material [10,11].

As a consequence, industrial UV nanosecond lasers represent a relatively low-cost and effective tool for machining a wide variety of materials, in particular polymers [12], glass [13,14], and transparent dielectrics [15]. Cutting, drilling, engraving, and structuring of both polymers and glass has been widely reported [12,13,14,15], leading to applications in a variety of industrial sectors such as portable electronics [16], semiconductors [17,18], and biomedical applications [19], where high-precision and high-quality machining are required. Industrial applications sparked the ongoing trend to improve laser stability, reduce the cost per watt, and increase the throughput [20]. The latter can be achieved by increasing the pulse energy, the repetition rate, or both. An increase in average power is another way to increase the throughput and is currently addressed by improving the performances of nonlinear crystals used for a third harmonic generation [21,22] and by engineering novel laser cavity architectures [23,24]. For instance, the use of rod-type fibers [25] enables the implementation of industrial lasers delivering high-quality Gaussian beams in UV ns pulses of several tens of µJ with a repetition rate in the MHz regime.

Increased throughput depends on the scanning system as well as the laser source. The fastest scanners, e.g., polygon and resonant, sacrifice flexibility for high-speed linear scanning [26]. On the other hand, galvo scanners have limited performance due to mechanical issues and rely on fast acceleration at the beginning of a scanning line and deceleration at its end. When reaching repetition rates in the MHz range, the accuracy and precision in pulse deposition are reduced, leading to nonhomogeneous structuring or local over-machining, observed particularly during scanner acceleration or deceleration. The universality of this issue led to technological solutions being introduced, e.g., the scanning strategy called skywriting and research related to its optimization [27,28]. The skywriting approach inherently increases the processing time with additional scanner motion involved, in turn leading to a reduced throughput. Maximal scanner throughput can theoretically be reached by controlling the pulse emission through the so-called pulse-on-demand (POD) approach. The POD operation refers to the laser source delivering pulses in sync with an external trigger whose frequency does not exceed the maximal available laser repetition rate [29,30,31,32,33]. To stabilize the output pulse energy, the laser resonator and/or amplification stages must be kept in equilibrium. In our POD operation realization, the equilibrium was achieved by incorporating two different seed sources that were separated at laser output by polarization/wavelength/peak power filtering. One source was turned on only for the demanded pulses, while the other kept the resonator and amplifiers in equilibrium while there was no demand for output laser pulses.

In this work, we present a novel POD module utilized jointly with an MHz rod-type UV nanosecond laser. To the best of our knowledge, an all-fiber UV nanosecond laser operating in an intrinsic POD regime, i.e., realized without an external pulse picker, has been realized for the first time. The laser was coupled with fast galvanometric scanners, achieving scanning speed values of up to 20 m/s on the material. We evaluated the performance of the setup in terms of precision, accuracy, and stability. We report on a comprehensive set of application-oriented experiments, showing a critical increase in the accuracy of pulse-to-pulse deposition compared to the standard operation regime while overcoming the skywriting throughput limitation. We studied the effects of scanning speed on POD operation, single- and multipass laser percussion drilling performance of metal and polymer materials, surface structuring of FTO (Fluorine-doped Tin Oxide)-glass sandwich material, and ablation efficiency for pulse durations in the range of 1–4 ns.

## 2. Materials and Methods

The experimental setup was designed as an open bench processing system built with industrial-grade equipment typically incorporated with laser-based material processing. The UV laser beam (wavelength approx. 343 nm) was guided to x-y galvo scanners (ScanLab Excelliscan 14) and focused through a 100 mm f-theta lens (Figure 1A), resulting in a maximal scan speed of 20 m/s. The calculated 1/e^2^ laser spot diameter on the material was 11 ± 1 µm, yielding a maximal pulse fluence of about 15 J/cm^2^. The laser beam was focused on the sample’s surface, and a 3D linear stages system was used for precise sample positioning.

The laser was based on a direct-modulation diode system for seeding, followed by multiple amplifier stages and final conversion into UV. The POD-enabled seeding stage was custom designed for the experiments and combined with the amplifying stage provided by Bloom Lasers. The UV output laser beam exhibits a high beam quality factor, expressed by the M-squared parameter M^2^ < 1.1. The amplifying stage enables output powers up to 30 W at 400 kHz repetition rate or up to 300 μJ pulse energy while keeping a high-quality output beam (astigmatism below 8% and ellipticity below 5%). For the microstructuring experiments, the output UV nanosecond pulses were set between 1.0 ns and 4.0 ns in duration, with average power reaching around 15 W at a 1.0 MHz repetition rate measured at the laser output and a corresponding 15 μJ pulse energy. The stable POD operation was ensured by using an additional seeding diode, called an idler diode, coupled into the same amplifier chain. This enabled stable conditions along the amplifier chain, achieved by idler pulses emitted instead of signal pulses during periods of no laser output demand. Idler pulses were, in turn, filtered out by the third harmonic generation process.

Scan vectors were generated in a proprietary CAD software from ScanLab and sent to the scanner. In return, the galvo scanners provided a train of trigger signals for pulse-on-demand operation during each scan vector. The whole POD signal for a 15 mm long line is shown in Figure 1B, with the zoomed-in panel providing a closer look at the first 12 trigger pulses. For a straight scanning line with a target scanning velocity of 20 m/s, the resulting laser output frequency varied from approx. 30 kHz at the line’s beginning to 1.0 MHz in the middle of the scanning line, where scanners finished the acceleration and kept a constant top speed. The laser was set to output pulses at frequencies up to 1.0 MHz, with a temporal resolution of 5 ns. This temporal resolution is defined by the ±2.5 ns timing jitter, which is negligible for the microstructuring application, even at the highest scanning speeds. The latency delay between the scanner’s output POD signal and the pulse emission was compensated for by the laser.

Laser-made microstructuring was applied to various industrial-standard materials, commonly used in combination with ablation-based laser processing. UV nanosecond pulse energy was set to sufficient levels to induce material evaporation and plasma formation, as well as melt ejection. The following materials and approaches were used:i.Evaluation of POD in comparison with standard and skywriting scanning regimes in terms of structure quality and precision were carried out on Kapton and polished stainless steel, scanning squares at different scanning speeds and at a fixed pulse-to-pulse distance. The materials were chosen to ensure smooth surface finishes and obtain results on two optically entirely different materials (dielectric and metal, respectively).ii.The setup accuracy was tested via precision ablation of the ITO layer on glass substrate, comparing the actual ablation crater positioning with the set values at the highest scanning speed at varying pulse-to-pulse distances.iii.The surface ablation experiments, a microstructuring example, were conducted on polished stainless steel, enabling a comparison of structure depth and bottom surface properties at different pulse durations, as well as measuring the edge roughness achieved using the POD regime.

The results section is organized into subsections according to the division of experimental approaches. Findings on precision, accuracy, and ablation properties of the novel laser processing setup are presented in the specified order.

Measurements of resulting structures were carried out using a high magnification optical microscope in both bright field and dark field modes (Olympus BX53M microscope with Olympus MPlanFL 10×, 20×, and 50× objectives). High magnification enabled precise focal plane recognition for measurements of structure depth with high vertical sensitivity. Single craters in the ITO layer were imaged by bright field microscopy and analyzed by a custom Matlab feature recognition script, typically averaging 10–20 craters at each scanner’s setting, to obtain the system precision results in Section 3.2. Additional surface analysis was performed using a scanning electron microscope (SEM) VEGA3 by TESCAN, allowing images at higher magnification (up to 1000 times) compared to optical microscopy. Possible chemical and structural material changes due to laser irradiation [34,35,36] were not imaged nor expected to differ compared to existing literature.

## 3. Results

### 3.1. Scanning Speed Effects

The two most common scanning strategies, i.e., conventional scanning without inertial effects compensation and skywriting, are compared to the POD-based approach. Figure 1 presents a direct comparison of structure quality obtained using different scanning strategies. Isolated ablation craters on a stainless steel surface near the corner of a square scanning path were chosen as a representative shape to highlight the key differences between the strategies. The intercrater distance was fixed at 20 μm, and the scanning speed was increased from 0.25 m/s to 20 m/s, with the lowest speed positioned innermost and increasing outwards:$$v=\left\{0.25;0.50;0.75;1.0;1.5;2.0;2.5;3.0;5.0;7.5;10;15;20\right\}\;m/s.$$

Both single-pass scans and *N*-pass scans (*N*—number of passes) are presented for each scanning strategy. The conventional scanning strategy, using a fixed laser repetition rate and standard vector scanning, results in evenly spaced craters only for the lowest scanning speed (Figure 2A,D, *v* = 0.25 m/s). At higher scanning speeds, a reduction in intercrater distance is observed during the acceleration/deceleration period of the scanner. Further increasing scanning speeds cause crater accumulation at increasingly long distances from the corner. As the craters start to overlap during the acceleration period, over-machining around the corner is observed. This effect is more pronounced in the case of *N* = 10 (Figure 2D).

The use of skywriting makes it possible to keep intercrater distance constant for all scanning speeds for both single-pass processing and *N* = 10, shown in Figure 2B,E. Another finding shown in Figure 2B,E is that the on/off signals need to be precisely synchronized with scanner motions in order to begin the scanning line at the correct position, and a fixed repetition rate results in positioning the jitter of the first pulse in a line. For example, at 20 m/s and 1.0 MHz laser repetition rate, the positioning of the first pulse is only precise to within 20 µm. These effects are shown in the outermost corners in Figure 2C,D.

The third option, the POD strategy, makes it possible to use standard vector scanning and ensure full throughput at undiminished scanning speeds, acceleration, or scanned distances. Intercrater distances remain constant under all conditions, as shown in Figure 2E,F. As the laser repetition rate is constantly adjusted to the scanner motion, the achieved accuracy depends only on the combination of laser timing response and scanner precision. The accuracy of our experimental system is evaluated in Figure 3.

### 3.2. Positioning Accuracy Measurement

The setup positioning accuracy measurement graph is presented in Figure 3, showing the system accuracy in terms of crater position deviation from the set value. The average deviation of the measured distance between two successive craters on material from the set value in the galvo scanner software is shown for different intercrater distances, including the standard deviation measured over 10–20 crater distances. The scanning speed was kept constant at 20 m/s. The laser repetition rate was set accordingly to achieve set distances from 20 µm to 50 µm, resulting in repetition rates between 1.0 MHz and 400 kHz, respectively. The highest scanning speed corresponds to the worst-case scenario, where the expected precision of the system is at its lowest. The graph in Figure 3A shows no observable trends, either in absolute deviation value or in standard deviation amplitude, with the final accuracy being independent of the intercrater distance. The measurement itself is influenced by impurities in the material, causing a variation in crater shape and size. Nevertheless, the precision of our setup is estimated to be within ±0.5 µm of the target, which is consistent with previously reported data [28].

A stable intercrater distance is important to successfully execute fast and repetitive scan lines without damaging either the substrate or the surrounding material. To demonstrate the principle, we conducted multipass scans on ITO material on glass. Figure 3B shows isolating lines in ITO, realized with *N* = 1–10 passes at a fixed scanning speed and pulse energy (6.0 m/s and 1.5 μJ, respectively). A single pass at the given laser and scanner settings is not enough to selectively remove the entire thickness of the ITO. After two passes, some residual ITO is still present, while 5 and 10 passes result in isolating lines of similar width and quality. The isolation properties achieved were tested using a multimeter to measure surface resistivity.

The system relative precision was evaluated via the multipulse percussion drilling approach, probing the relative laser pulse positioning precision. The experiment was based on a laser drilling strategy that minimizes material overheating, applied to the processing of stainless steel. The laser processing was followed by the analysis of spot evolution over multiple pulse repetitions on a single spot. First, we observed single-pulse ablation craters on the stainless steel surface and compared the results with a simple model for pulsed laser ablation, typically valid for ultrashort pulses [4] but also applicable in cases where heat transfer during the pulse duration is negligible [37]. The expression was established for laser processing by Liu et al. [38] and adopted in research works since Furmanski et al. [39]. The expression is derived from the Gaussian spatial intensity distribution of the pulse and basic ablation properties. The model predicts the ablation crater diameter scaling as a function of the pulse energy *E_p_*:(1)$$d_{c}\left(E_{p}\right)=\sqrt{2w_{0}^{2}\ln\left(\frac{E_{p}}{E_{th}}\right)}.$$

The parameters are laser spot size radius $w_{0}$ and threshold pulse energy for ablation $E_{th}$. In case of multiple pulses per spot and no incubation effects [9] due to long times between subsequent pulses on the same crater position, a first approximation can be made that $E_{tot}=NE_{p}$, where *N* is the number of pulses per spot, leading to an increased crater size at a fixed pulse energy:(2)$$d_{c}\left(N\right)=\sqrt{2w_{0}^{2}\ln\left(\frac{NE_{p}}{E_{th}}\right)}.$$

The laser spot size radius $w_{0}$ and the coefficient $E_{p}/E_{th}$ were taken as free parameters for the fit. The agreement between the measured data and the simple model function is rather good for stainless steel ablation (R^2^ = 0.98). The craters in the stainless steel material for *N* = {10,20,50,100} pulses are shown in Figure 4A–D, and the corresponding diameters and fitted model functions are on the graph in Figure 4. For Kapton, the simple approximation does not hold to the same degree (R^2^ = 0.69).

### 3.3. Microstructuring and Surface Quality

Microstructures in stainless steel were chosen for the analysis of surface morphology and ablation depth as functions of both the pulse duration and pulse energy. An established approach for microstructuring was used, called milling, to create a relatively large structure in lateral dimension (2 × 2 mm^2^) in order to isolate the effects of the edges from the surface processing. To obtain measurable structure depths at all pulse energies, the following structuring parameters were used: 100 repetitions using x-y hatch with 4.0 μm pulse-to-pulse and line-to-line distances.

The graph of optically measured structure depths versus pulse energy for three pulse durations is shown in Figure 5. We found that structure depth increases with pulse duration while surface quality decreases. A comparison is presented in Figure 5, with panels A–D showing the bottom morphology for four different process parameter combinations. Starting from low pulse energies, the bottom roughness was below 1 μm for all pulse durations. At higher pulse energies, the ablation regime experiences a transition to a significantly rougher structure bottom, with bottom roughness reaching up to 10 μm. This transition happens for all observed pulse durations but at significantly different pulse energies. For the 1 ns pulse duration, this transition happens between 8 μJ and 12 μJ pulse energies (Figure 5A,B), and for the 4 ns pulse duration, it occurs between 1 μJ and 2 μJ pulse energies (Figure 5C,D). The most interesting result of the comparison is that the resulting processing efficiency is much higher when using shorter pulses and constraining the processing to the low roughness regime. For example, the structure depth corresponding to Figure 5A was 38 μm and only 7 μm for the parameters used for the structure shown in Figure 5C.

Detailed analysis of structure bottoms and edges for different pulse durations and pulse energies was carried out based on SEM imaging. Figure 6 shows detailed views of the structures for three different laser parameter sets: A–1 ns pulse duration, 3 μJ pulse energy; B–2 ns pulse duration, 2 μJ pulse energy; C–4 ns pulse duration, 1 μJ pulse energy. Three different observations were made:Structure edges are steep and clear of debris, with no evidence of over-machining, pointing to optimal operation of the POD laser in combination with the scanners. Edge straightness was measured to be better than ±1 μm for all laser parameters shown, with some visual deviations caused by uneven material ejection.Bottom roughness is similar for all three parameter sets, with a distinctive change in morphology using 2 ns pulses, where different periodicity is observed compared to the other two cases.The shortest, 1 ns pulses, are the least efficient at microstructuring (comparison shown in Figure 5, the data for 1 ns pulses are bottommost throughout the graph), and the resulting bottom morphology remains the least rough at higher microstructure depths.

For all pulse durations, a certain amount of melting is observed during a single pulse interaction with the steel. In Figure 6B, melt formation sustained over a few neighboring pulses is most likely the cause of the typical morphology observed. The sustained melt formation on a larger scale occurs at higher pulse energies, leading to bottom morphology shown in Figure 5B,D.

## 4. Discussion

A first-time demonstration of POD operation in a high-power UV nanosecond laser has been realized and tested on relevant microstructuring applications. First, three different approaches to galvanometric-based scanning were tested and compared, namely skywriting and POD, compared to conventional scanning. The results are in line with previously reported work on femtosecond POD technology associated with the same fast galvo scanner for laser micromachining, the latter showing clear benefits of POD in both throughput and quality.

Further, we evaluated the accuracy and precision of the experimental laser microstructuring setup by separately measuring the absolute and relative laser pulse positioning. The absolute setup accuracy is determined by numerous factors, including scanner precision, laser emission timing jitter, sample positioning precision, and other less predictable effects, such as the response to external vibrations. We have separately measured the timing jitter of the laser itself, which was within ±2.5 ns from the ideal timing at a fixed delay after the input trigger. The remaining factors together contributed to the overall absolute system accuracy being within ±0.5 µm of the target, again consistent with previously reported data. For the relative precision measurement, multipulse percussion drilling was used. The crater evolution trend observed is indicative of the relative precision; in case of drifts or instability, the crater gets elliptical or extends with the increasing number of pulses. If the relative precision is comparable to one-tenth of the beam diameter or better, the crater diameter slowly saturates with the number of pulses, extending just beyond the entire beam diameter, with the precision-related deformation indistinguishable. Our findings confirm the latter behavior and thus put the relative precision in line with absolute accuracy.

Lastly, the microstructuring capability was investigated. Both the throughput, measured in terms of structure depth and the resulting surface roughness, were investigated. The research indicates that on stainless steel, the use of longer, 4 ns pulses is more efficient but results in significantly increased surface roughness at high pulses energies, more than 10 times greater compared to low pulse energies. When comparing structures with similar surface roughness, the use of shorter, 1 ns pulses resulted in a higher throughput/bigger structure depth at equal total energy density, i.e., offering a better compromise between efficiency and surface quality, which was also confirmed using SEM imaging. Further research is needed to evaluate a broader range of pulse durations and to study the ablation efficiency evolution with energy in detail on various materials.

The POD operation combined with a high-power UV nanosecond laser output is important for applications requiring optimized high throughput, e.g., PCB via drilling, cutting, and depaneling, or metal surface texturing for batteries, as well as for applications benefiting from a precisely controlled energy distribution on material, e.g., ITO patterning or wafer scribing.

## Figures and Tables

**Figure 1 micromachines-14-00843-f001:**
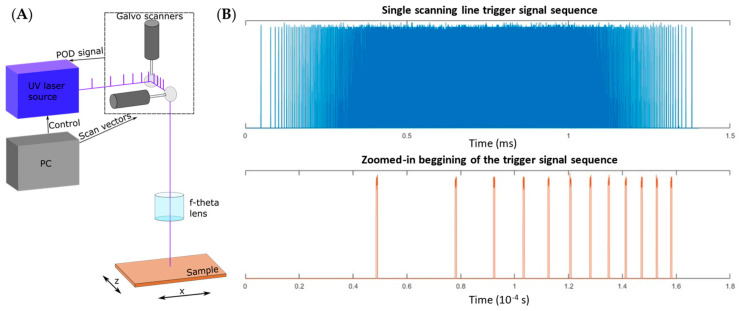
(**A**) Experimental setup schematics. (**B**) POD sequence received from the galvo scanners, based on scanning a 15 mm long straight line at maximal velocity of 20 m/s. Lower frequencies at the beginning and the end correspond to scanner acceleration periods.

**Figure 2 micromachines-14-00843-f002:**
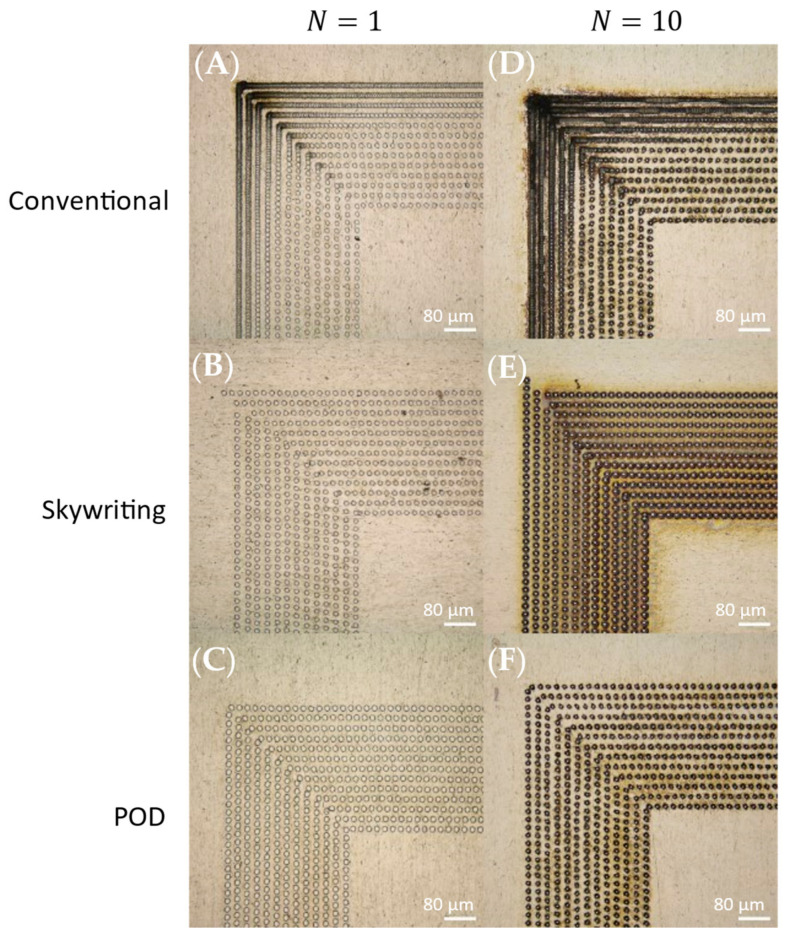
Comparison of scanning strategies by effects in corners of scanning lines—ablation craters in stainless steel for a set of different scanning velocities. The scanning speeds used were *v* = {0.25;0.50;0.75;1.0;1.5;2.0;2.5;3.0;5.0;7.5;10;15;20} m/s, with the lowest speed positioned innermost and increasing outwards. Scale bars are equal to 80 μm.

**Figure 3 micromachines-14-00843-f003:**
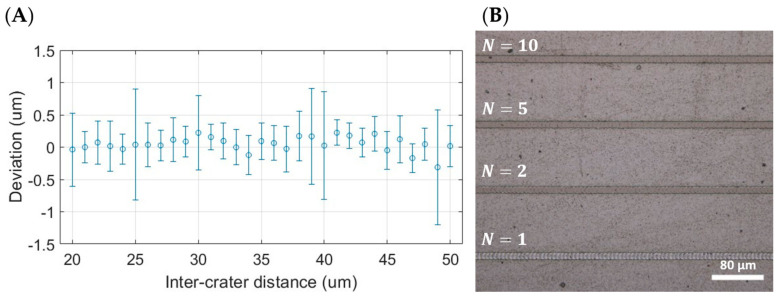
(**A**) Graph showing the deviation from the expected intercrater distance at various intercrater distances and at maximal scanning speed (20 m/s). (**B**) Examples of ITO isolation using multiple high-speed scanning passes and an optimized fluence to achieve clean lines with minimal effects on the surrounding material. Scale bar is equal to 80 μm.

**Figure 4 micromachines-14-00843-f004:**
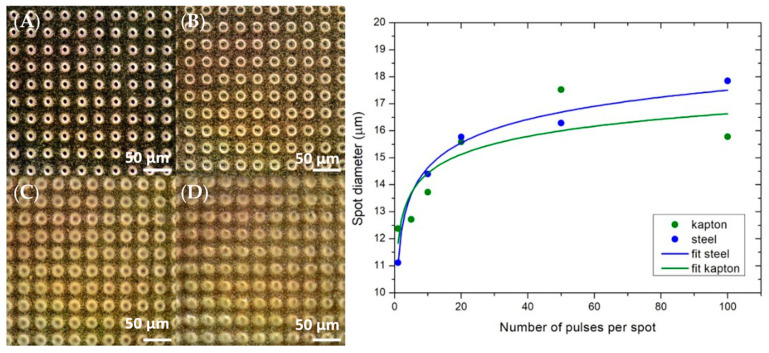
Panels (**A**–**D**) show arrays of craters from multipulse ablation experiments, for *N* = {10,20,50,100}. The crater edges become fuzzy on panels (**C**) and (**D**) due to high optical magnification and the focus position kept on the material’s surface. Scale bars are all equal to 50 μm. The graph shows averaged data of spot diameter evolution with the number of pulses per spot (*N*) and added best function fits.

**Figure 5 micromachines-14-00843-f005:**
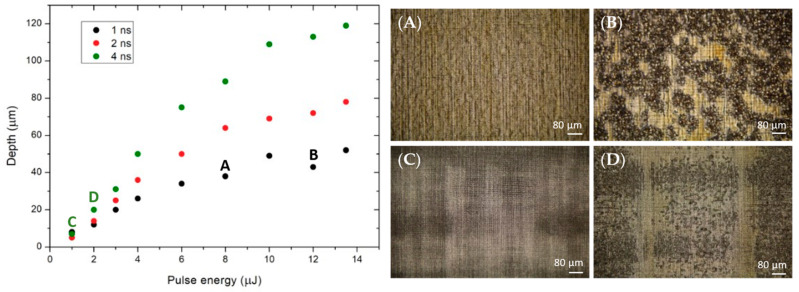
Structure depth evolution graph as a function of the laser pulse energy for three different pulse durations used in the experiment. Panels (**A**–**D**) show bottom morphology for four different parameter combinations, indicated on the graph (A: 1 ns, 8 μJ; B: 1 ns, 12 μJ; C: 4 ns, 1 μJ; D: 4 ns, 2 μJ). Scale bars are all equal to 80 μm.

**Figure 6 micromachines-14-00843-f006:**
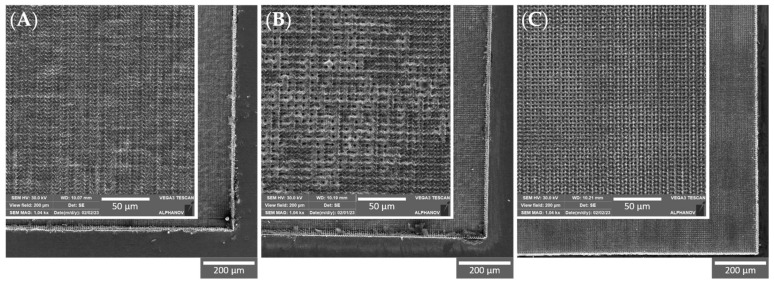
Laser surface structuring of stainless steel material. Insets show close-up views of structure bottoms, while the larger images allow for structure edge evaluation. Laser parameters shown are the following: (**A**) 1 ns pulse duration, 3 μJ pulse energy; (**B**) 2 ns pulse duration, 2 μJ pulse energy; (**C**) 4 ns pulse duration, 1 μJ pulse energy. Structure depth decreases from A to C, while bottom roughness is smallest for A.

## Data Availability

The data presented in this study are available on request from the corresponding author. The data are not publicly available.

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
