# Peer review of "Pulse-on-Demand Operation for Precise High-Speed UV Laser Microstructuring"

_micromachines, 2023, doi:10.3390/mi14040843_

Round 1

Reviewer 1 Report

In the article “Pulse-on-demand operation for precise high-speed UV laser microstructuring”, the authors investigated the microstructuring by using high-power UV nanosecond laser in detail. The authors well analyzed the mechanism using material analyses, experimentally and theoretically. However, this article is hard to understand the contents at once. I cannot recommend publication at “Micromachines” in this stage. The authors need to carry out “minor revisions”.

1.     The authors should enlarge the font size of Fig.1 and Fig.6

2.     According to laser irradiation, there seems to be a material change such as a phase change of a material or a segmentation, but there is no analysis. Please check this by analyzing other substances such as XRD, EDS, and TEM in addition to SEM.

3.     It is necessary to explain in more detail the mechanism in which the material is ablated by the laser.

4.     What applications can be applied when using this type of process? It is necessary to specify this in the introduction and conclusion.

5.     There are various examples of changing the structure of a material using light and applying it to a new application, and I think the impact of the paper will be increased if it is included as a reference in the introduction.

Small 15.48 (2019): 1901529.

Sensors and Actuators B: Chemical 348 (2021): 130714.

Author Response

Point-by-point response to the reviewer’s comments

Reviewer: 

Comments and Suggestions for Authors

In the article “Pulse-on-demand operation for precise high-speed UV laser microstructuring”, the authors investigated the microstructuring by using high-power UV nanosecond laser in detail. The authors well analyzed the mechanism using material analyses, experimentally and theoretically. However, this article is hard to understand the contents at once. I cannot recommend publication at “Micromachines” in this stage. The authors need to carry out “minor revisions”.

Response: Thank you for your review. Please find the point-by-point replies below in italics.

  1. The authors should enlarge the font size of Fig.1 and Fig.6

Response 1: We have increased the font size in Figure 1 and Figure 6.

  1. According to laser irradiation, there seems to be a material change such as a phase change of a material or a segmentation, but there is no analysis. Please check this by analyzing other substances such as XRD, EDS, and TEM in addition to SEM.

Response 2: While indeed laser irradiation causes phase changes, oxidation, and other material modifications, we believe that the general interaction of a nanosecond (UV) laser pulse with materials used in this work has already been studied (references 7-19) and thus, further analysis of samples using XRD, EDS, and TEM would not bring a significant improvement to the work.

  1. It is necessary to explain in more detail the mechanism in which the material is ablated by the laser.

Response 3: We have added a more detailed description of the nanosecond ablation process, around line 120 of the manuscript.

  1. What applications can be applied when using this type of process? It is necessary to specify this in the introduction and conclusion.

Response 4: The text in Introduction, lines 39-42, already describes and references relevant industrial applications. We have added a mention of the envisioned applications to the end of Conclusion.

  1. There are various examples of changing the structure of a material using light and applying it to a new application, and I think the impact of the paper will be increased if it is included as a reference in the introduction.

Small 15.48 (2019): 1901529.

Sensors and Actuators B: Chemical 348 (2021): 130714.

Response 5: We have added the references to better reflect the possible material changes induced by the laser-material interaction.

Reviewer 2 Report

In the figure captions, values for the scales are mentioned. In my opinion, it would be easier to include that size information directly in the images.

Author Response

Point-by-point response to the reviewer’s comments

Reviewer:

In the figure captions, values for the scales are mentioned. In my opinion, it would be easier to include that size information directly in the images.

Response: Thank you for your kind review. We have improved images by including scale bar dimensions directly on them where relevant.